# Endophytic Colonization by *Beauveria bassiana* and *Metarhizium anisopliae* in Maize Plants Affects the Fitness of *Spodoptera frugiperda* (Lepidoptera: Noctuidae)

**DOI:** 10.3390/microorganisms11041067

**Published:** 2023-04-19

**Authors:** Nimra Altaf, Muhammad Irfan Ullah, Muhammad Afzal, Muhammad Arshad, Sajjad Ali, Muhammad Rizwan, Laila A. Al-Shuraym, Seham Sater Alhelaify, Samy Sayed

**Affiliations:** 1Department of Entomology, University of Sargodha, Sargodha 40100, Pakistan; altaf.nimra@gmail.com (N.A.); chafzal64@yahoo.com (M.A.); makuaf@gmail.com (M.A.); 2Baba Guru Nanak University, Nankana Sahib 39100, Pakistan; 3Department of Entomology, The Islamia University of Bahawalpur, Bahawalpur 63100, Pakistan; sajjad.ali@iub.edu.pk; 4Beekeeping and Hill Fruit Pests Research Station, Rawalpindi 46300, Pakistan; ranarizwanjabbar@yahoo.com; 5Department of Biology, College of Science, Princess Nourah Bint Abdulrahman University, P.O. Box 84428, Riyadh 11671, Saudi Arabia; laalshuraym@pnu.edu.sa; 6Department of Biotechnology, Faculty of Science, Taif University, P.O. Box 11099, Taif 21944, Saudi Arabia; biotechnology31@hotmail.com; 7Department of Economic Entomology and Pesticides, Faculty of Agriculture, Cairo University, Giza 12613, Egypt; 8Department of Science and Technology, University College-Ranyah, Taif University, P.O. Box 11099, Taif 21944, Saudi Arabia

**Keywords:** biology, colonization, entomopathogenic fungi, endophyte, *Spodoptera frugiperda*

## Abstract

The fall armyworm, *Spodoptera frugiperda* (Noctuidae; Lepidoptera), is a serious threat to food security as it has the potential to feed on over 353 plant species. To control this insect pest, endophytic colonization of entomopathogenic fungi (EPF) in plants is being considered as a safer and more effective alternative. This study evaluated the efficacy of two EPFs, *Beauveria bassiana* and *Metarhizium anisopliae*, for endophytic colonization using foliar spray and seed treatment methods on maize plants, and their impact on the survival, development, and fecundity of *S. frugiperda*. Both EPF effectively colonized the maize plants with foliar spray and seed treatment methods, resulting in 72–80% and 50–60% colonization rates, respectively, 14 days after inoculation. The EPF negatively impacted the development and fecundity of *S. frugiperda*. Larvae feeding on EPF-inoculated leaves had slower development (21.21 d for *M. anisopliae* and 20.64 d for *B. bassiana*) than the control treatment (20.27 d). The fecundity rate was also significantly reduced to 260.0–290.1 eggs/female with both EPF applications compared with the control treatment (435.6 eggs/female). Age-stage-specific parameters showed lower fecundity, life expectancy, and survival of *S. frugiperda* when they fed on both EPF-inoculated leaves compared with untreated leaves. Furthermore, both EPFs had a significant effect on population parameters such as intrinsic (*r* = 0.127 d^−1^ for *B. bassiana*, and *r* = 0.125 d^−1^ for *M. anisopliae*) and finite rate (*λ* = 1.135 d^−1^ for *B. bassiana*, and *λ* = 1.1333 d^−1^ for *M. anisopliae*) of *S. frugiperda* compared with the control (*r* = 0.133 d^−1^ and *λ* = 1.146 d^−1^). These findings suggest that EPF can be effectively used for the endophytic colonization of maize plants to control *S. frugiperda*. Therefore, these EPFs should be integrated into pest management programs for this pest.

## 1. Introduction

The fall armyworm, *Spodoptera frugiperda* (Noctuidae; Lepidoptera), is an invasive insect pest that attacks various economically important crops. It is native to tropical and subtropical regions of the Americas and has rapidly spread to other countries worldwide due to its strong flying capacity and migratory behavior [1]. This pest has a polyphagous feeding habit and is known to feed on over 353 plant species, with corn being its most preferred host, causing significant annual losses in corn production [2]. *S. frugiperda*’s extensive feeding on economically important crops is increasingly threatening agricultural productivity and exacerbating food insecurity [3]. Since its introduction to Pakistan in 2019 [4], *S. frugiperda* has caused significant damage to the maize crop, leading to substantial losses.

Farmers commonly rely on synthetic insecticides to control insect pests in their fields, but this approach can lead to a host of problems, such as insect resistance, harm to non-target organisms, and environmental damage [5]. Consequently, alternative strategies for pest management are needed. One such strategy is microbial control, which has proven to be highly effective against insect pests [6,7]. Entomopathogenic fungi (EPF) are particularly promising for integrated pest management, as they are cost-effective and have no harmful effects on humans or the environment [8,9]. There are around 750 known EPF species that infect various insects and mites, each with its own specific target [8]. The genera *Beauveria*, and *Metarhizium* are especially effective against lepidopterous insect pests [10]. In addition to their use as biological insecticides, many EPF species are capable of colonizing plant tissues [11,12]. Although only a few EPF species occur naturally as endophytes, numerous successful attempts have been made to introduce various EPFs into plants using different techniques [12]. This endophytic colonization of EPF can help improve plant growth and reduce pest densities in a variety of economically important crops [13,14,15,16].

The ability of *B. bassiana* to colonize maize plants and produce secondary plant metabolites that infect herbivorous insects is considered highly effective [17,18]. Endophytically colonized entomopathogenic fungi have been recovered from different parts of plants, such as leaves, stems, and roots, and these colonized plants show high virulence against insect herbivory [19,20]. *M. anisopliae* has been introduced as an endophyte in several plants, including tomato, cassava, and oilseed rape, with negative effects on the larvae of *Plutella xylostella* [21,22,23]. The insecticidal effect of such endophytic EPF colonization on major plant insect pests can be useful in IPM strategies. The main objective of our study was to evaluate the endophytic effect of *B. bassiana* and *M. anisopliae* on the biology and survival of *S. frugiperda*.

## 2. Materials and Methods

### 2.1. Insect Culture

The eggs and larvae of *S. frugiperda* were obtained from a maize field located at the research farm (32°07′57.3″ N 72°41′30.2″ E) of the University of Sargodha. The culture was maintained under controlled conditions of 65 ± 5% relative humidity and 27 ± 2 °C at the Biocontrol laboratory of the Entomology Department at the University of Sargodha. Neonate larvae were fed an artificial diet prepared using the method suggested by Sorour et al. [24]. The adults were moved to plastic cages (30 × 30 × 30 cm) and provided with a 10% sugar solution for food. Muslin cloth was provided in plastic jars to facilitate oviposition. The F3 generation was used for further experiments.

### 2.2. Plant Culture

The researchers purchased hybrid maize seeds (*Zea mays* L.; var. HY-CORN 11 Plus, ICI Pakistan Ltd., Lahore, Pakistan) from a local market in Sargodha. The seeds were sterilized by soaking them in a 70% ethanol solution for two minutes. The seeds were washed with 1.0% sodium hypochlorite (DAEJUNG Chemicals & Metals Co., Ltd., Gyeonggi-do, Republic of Korea) for 2 min followed by three times washing with distilled water, after which they were soaked in distilled water at 4 °C for 24 h before planting. The seeds were then sown in plastic pots (11 × 12 cm) containing a mixture of soil, perlite, and vermiculite in equal proportions (1:1:1), and the planting medium was autoclaved three times for 45 min at 121 °C with an interval of 24 h between each autoclave. The plants were grown in a greenhouse and irrigated as needed, without the application of any pesticides or fertilizers throughout the experiment.

### 2.3. Entomopathogenic Fungi

Entomopathogenic fungi, *B. bassiana* and *M. anisopliae* were obtained from AgriLife SOM Phytopharma (Hyderabad, Telangana. India) Limited in talc form [25]. The conidial spore suspension for both fungi was adjusted to 1 × 10^8^ conidia mL^−1^ by using Neubauer hemocytometer [26]. A germination test [27] was performed for both fungi to evaluate the viability of conidial spores. The conidial suspensions with ≥90% germination were used for plant inoculation.

### 2.4. Plant Inoculation with Entomopathogenic Fungi by Foliar Application

Maize seeds that had been sterilized were planted in pots containing a sterile planting medium, as described earlier. When the maize seedlings were three weeks old at the growth stage BBCH 15 (5 leaves unfolded) [28], they were sprayed using a hand sprayer with an average of 3 mL of spore suspensions of each fungus in distilled water with 0.01% Tween 80. In the control treatment, plants were sprayed with 3 mL of a solution consisting of distilled water and 0.01% Tween 80. Each treatment was sprayed directly onto the leaves. To prevent conidial runoff, the surface of each pot was covered with aluminum foil while spraying. The experiment was repeated four times and 5 plants were selected randomly, totaling 20 plants for each treatment. Independent batches of plants and EPF were used in each treatment. In order to neutralize the effect of position, pots of each treatment were placed in a randomized complete block design (RCBD) in a greenhouse.

### 2.5. Plant Inoculation with Entomopathogenic Fungi through Seed Treatment

In this method, surface-sterilized maize seeds were dipped in 10 mL of conidial spore suspension of each fungus for 24 h. A sterilized paper towel was used to dry the seeds for 30 min prior to sowing in pots containing the sterile planting medium as discussed above. In the control treatment, seeds were soaked in distilled water with 0.01% Tween 80 solution for 24 h prior to sowing. The same numbers of replications and designs were used as in the foliar application method.

### 2.6. Colonization of Plants by Endophytic Entomopathogenic Fungi

Leaf samples were collected 14 and 28 days after the inoculation of EPF. For each sampling day, ten plants were selected, and the fourth true leaf was taken from each plant for each treatment. The leaves were washed with distilled water, sterilized with 70% ethanol for 2 min, and then with 1.0% sodium hypochlorite for 2 min. The samples were then rinsed twice with sterile distilled water. Sterilized scalpels were used to slash the leaves into small pieces. Each piece of the leaf was plated individually on Potato Dextrose Agar (PDA) medium. On each sampling day, an average of four pieces of leaves were collected from each plant. The samples were placed in Petri plates containing 20 mL of PDA and incubated at 25 °C in the dark. The plates were observed after 7 and 15 days of PDA inoculation to record fungal growth. The percent colonization frequency was calculated using the following formula:$$CF=\frac{No.\;of\;plant\;pieces\;showing\;fungal\;growth}{Total\;no.\;of\;plated\;plant\;pieces}\times 100$$

### 2.7. Endophytic Effects of Entomopathogenic Fungi on Life Table Parameters of S. frugiperda

The most effective method of plant inoculation was determined based on the highest colonization rate of EPF (Figure 1). Highly colonized plants from the foliar spray method (highly effective) were used in the life table study. In each treatment, eighty 2-day-old first instar larvae were separated from the rearing colony and placed in Petri plates (one per plate). Treated maize leaves were provided as needed until pupation. In the control group, non-inoculated plant leaves were provided. The developmental period of each stage and survival rate were recorded daily. After pupal formation, all pupae were placed in separate Petri plates lined with cotton, and the pupal period was recorded. Adults from each treatment were paired and released into transparent plastic boxes (30 × 30 × 30 cm) with a honey solution provided as food. A healthy potted plant was placed in each cage, and muslin cloth was hung in the plastic boxes to facilitate oviposition. Newly laid eggs were transferred to Petri plates, and the total numbers of eggs were recorded daily. This experiment was conducted under controlled conditions at 25 ± 1 °C, 60–70% relative humidity, and a 16:8 h (light: dark) photoperiod. The life stages, including the egg incubation period, duration of each larval stage, total larval development time, pupal duration, pupal emergence into adults (females and males), the number of eggs laid by each female, and adult life were recorded.

### 2.8. Statistical Analyses

For percent colonization, data were analyzed by three-way ANOVA by keeping EPF, inoculation method, and time interval as main factors. Means were separated by LSD all-pairwise comparison test at a 5% level of significance. The development duration and survival rate from raw data were analyzed using age-stage, two-sex life table procedures using the TWO SEX-MS Chart program [29]. For the calculation of standard error, bootstrapping method (with 100,000 random samplings) was used by using the MS Chart program.

## 3. Results

Before being inoculated in maize plants, the viability of two entomopathogenic fungi, *B. bassiana* and *M. anisopliae*, was assessed on PDA plates. Both fungi had a germination rate of over 90% and were successfully inoculated in the maize plants. The frequency of endophytic colonization (CF) by *B. bassiana* and *M. anisopliae* varied significantly (F = 5.78, *p* < 0.05) depending on the inoculation method used. The highest CF percentage for both fungi was observed when using the foliar spray method compared with seed treatment. The colonization rate of both fungi was highest at 14 days after inoculation, compared with 28 days. At 14 days, the CF percentage of *M. anisopliae* was 80.0% using the foliar spray method and 65.0% using the seed inoculation method, while the CF percentage of *B. bassiana* was 72.5% when applied by foliar spray and 50.0% by seed inoculation method (Figure 1).

Table 1 presents the development period for each stage of *S. frugiperda* when feeding on leaves inoculated with EPF. The larval instars showed significant differences in their developmental period, except for L2 and L4 (*p* > 0.05). The larvae took 20.64 days to complete their developmental period when fed on *B. bassiana*-inoculated leaves, 21.21 days when fed on *M. anisopliae*-inoculated leaves, and 20.27 days when fed on untreated leaves. The pupal duration was extended to 8.20 days when *B. bassiana* was applied and 7.43 days when *M. anisopliae* was applied, compared to 6.91 days in the control. The longevity of female adults was longer than that of male adults in all treatments. However, when larvae were fed on *M. anisopliae*-inoculated leaves, female adult longevity was shorter (9.73 days) compared to adults in the control group (11.47 days) (Table 1).

The study found that the control group had a shorter adult pre-oviposition period (APOP) of 2.35 days, while the APOP period was longer in the *B. bassiana* and *M. anisopliae* treatments (2.67 days and 2.55 days, respectively). However, the total pre-oviposition period (TPOP) was longer in the control group (35.3 days) compared with the *B. bassiana* and *M. anisopliae* treatments (33.3 days). When immature stages were fed on EPF-inoculated leaves, the oviposition period of females was shorter (3.6–3.8 days) compared with the control group (4.58 days). The lowest fecundity rate was recorded in the *M. anisopliae* treatment (260.0 eggs/female), followed by 290.1 eggs/female in *B. bassiana*, compared with the control group (435.6 eggs/female). All reproductive parameters, including intrinsic increase rate (*r*) and finite increase rate (*λ*), net reproductive rate (*Ro*), and generation time (*T*) of *S. frugiperda*, were reduced in both EPF treatments compared with the control group (Table 2).

Figure 2 displays the age-stage-specific survival rate (*s_xj_*) of *S. frugiperda* after treatment with EPF. The curve represents the survival rate from the egg stage to age *x* and stage *j*. Male and female adults emerged on the 29th day in the control group and the 26th day in the *M. anisopliae* treatment group. In the *B. bassiana* group, the male emerged on the 28th day and the female on the 27th day (Figure 2). The life expectancy rate (*e_xj_*) curve shows the expected survival time of individuals of age *x* and stage *j*. The *e_xj_* curves of larvae and adults of *S. frugiperda* treated with both EPF were lower compared with the untreated (control) group. At age zero (*e*_01_), the *e_xj_* of *S. frugiperda* was 43.1 days in the control group, 33.5 days in the *B. bassiana* treatment group, and 30 days in the *M. anisopliae* treatment group (Figure 3). Females were predicted to live for 13.5 days and 13.7 days, while males were predicted to live for 11.5 days and 12 days when fed on maize plants inoculated with *B. bassiana* and *M. anisopliae*, respectively. In non-inoculated maize plants, females and males were predicted to live for 14.45 and 15.08 days (Figure 3).

Figure 4 displays the age-specific survival rate (*l_x_*), age-stage-specific fecundity (*f_xj_*), age-specific fecundity (*m_x_*), and age-specific maternity (*l_x_m_x_*) of *S. frugiperda* after EPF application. The fecundity rate of female *S. frugiperda* appeared on the 31st day in control, 29th day in *B. bassiana*, and on the 28th day in *M. anisopliae*. Overall, the maternity rate of *S. frugiperda* peaked on the 39th day in control and *B. bassiana*, and 43rd day in *M. anisopliae* (Figure 4). The age-stage reproductive value (*v_xj_*) indicates the future population growth of individuals of age *x* and stage *j*. At age zero (*v*_01_), the *v_xj_* of *S. frugiperda* was 1.146 d^−1^ in control, 1.135 d^−1^ in *B. bassiana*, and 1.133 d^−1^ in *M. anisopliae*. The highest reproductive value of female was observed in the case of control at age 36 days (*v*_36,9_ = 247.07 d^−1^). However, the *v_xj_* value was highest (*v*_33,9_ = 177.92 d^−1^) on the 33rd day in *B. bassiana*, and in the case of *M. anisopliae*, higher peaks of *v_xj_* were recorded; *v*_32,9_ = 154.39 d^−1^ at 32 days and *v*_34,9_ = 154.46 d^−1^ at 34 days (Figure 5).

## 4. Discussion

Entomopathogenic fungi (EPF) have been found to be effective in controlling several economic insect pests, providing an alternative to chemical control. However, unfavorable weather conditions may hinder the exposure of fungal spores in the field, thereby reducing their efficiency and level of utilization [30]. To address this, inoculating EPF as fungal endophytes can be a useful approach to reducing the negative effects of abiotic stressors [12], rather than relying on inundative methods. Previous studies have identified a variety of EPF as natural endophytes of important crops such as potato, maize, cotton, tomato, and chickpea [11,31,32,33]. Of the various EPF, *B. bassiana* and *M. anisopliae* are well-known for their ability to colonize plants endophytically [11]. In general, many recent investigations stated that numerous *B. bassiana* and *M. anisopliae* isolates have shown the high efficiency of these fungi in the infection and control of *S. frugiperda* larvae [34,35,36,37,38]. We conducted this study to investigate the endophytic effects of two different entomopathogenic fungi, *B. bassiana,* and *M. anisopliae*, using two inoculation methods: foliar spray and seed inoculation. The foliar spray method was found to be more effective in terms of high percent colonization in the leaves compared to seed inoculation. The choice of inoculation method may depend on the targeted plant part for endophytic colonization or the insect species to be controlled, such as sucking insects, root and stem borers, or leaf-chewing insects. According to [39], the foliar spray method is the easiest to use in the field. However, some studies have reported no colonization of EPF into the stem or leaf through seed inoculation [19,32], which could be due to the negative effects of microorganisms present in the soil that act as antagonists for EPF. The presence of both EPFs was higher 14 days after inoculation, but the percentage decreased on the 28th day. The percent colonization rate may depend on the fungal strains and plant species. The plant growth stage can also be another factor affecting the colonization rate of EPF. Rajab et al. [40] reported that the fungus was able to colonize cucumber plants more efficiently in the first stage of plant growth compared to the seedling stage. Rondot and Reineke [41] recorded the existence of *B. bassiana* in grapevine plants after 28 days of inoculation, while Akello et al. [42] reported it could be re-isolated up to 120 days after inoculation from banana plants. Posada et al. [39] isolated *B. bassiana* at low rates from coffee tissues after 120 days of inoculation.

Our study revealed that the larval and pupal stages were negatively impacted when feeding on leaves inoculated with EPF, particularly *M. anisopliae*. The developmental period was extended in larvae that fed on EPF-inoculated leaves compared to those that consumed untreated leaves. Our findings are consistent with previous research indicating that EPF can increase the developmental time of insects [43,44]. The prolonged development of immature stages of insects may be attributed to the reduced conversion of ingested and digested food after exposure to fungi, leading to slower larval development [43].

Our study found that feeding larvae on EPF-inoculated leaves, particularly *M. anisopliae*, adversely affected their larval and pupal periods. The developmental time was longer for those larvae fed on EPF-inoculated leaves than for those fed on untreated leaves. Our findings regarding the extended developmental time of insects due to EPF are consistent with previous studies [43,44]. This increase in developmental time could be due to a decrease in the conversion of digested and ingested food after fungal exposure, which slows the development of larvae [43].

The longevity of adults was reduced when their immature stages were fed on EPF-inoculated leaves. Similarly, the fecundity rate of female adults that emerged from surviving pupae fed on EPF-inoculated leaves was considerably reduced compared to the control. Other population parameters, such as *Ro*, *r*, *λ*, and *T*, were also reduced when using fungal endophytes. Therefore, inoculating plants with EPF can significantly reduce the feeding and oviposition of insect pests, as previously reported in studies on the bean stem maggot, *Ophiomyia phaseoli*, in bean plants [45] and the cotton leafworm, *S. littoralis*, in wheat plants [46]. Plants colonized by fungal endophytes exhibit feeding deterrence or antibiosis against their insect pests, which could be due to the synthesis of secondary metabolites by endophytic fungi. Plants colonized with EPF are less favorable to insects and indirectly affect the fitness of pests, as reported in previous studies [14,16,47,48,49,50]. Our findings are similar to previous studies showing the negative impact of endophytic fungi on the reproductive potential and lifespan of insects [51,52]. These negative effects could be due to secondary metabolites or the induction of a systemic response in the colonized plants [52]. The endophytic colonization of EPF in plants induces indirect detrimental impacts on target pests through various non-pathogenic mechanisms, including antixenosis, antibiosis, and induced systemic resistance [53]. The most commonly known endophytic fungi are *Beauveria* and *Metarhizium* spp., which can synthesize various secondary metabolites with antifungal, antibacterial, and insecticidal properties [54]. In this study, we did not evaluate the effect of these EPFs on the plant. However, Rajab et al. [40] reported no negative effects of *B. bassiana* colonization in cucumber plants on their pathogenicity. As an advantage, EPF can increase plant growth, as Rivas-Franco et al. [55] concluded that *Metarhizium* promoted maize vegetative growth. However, this depends on the EPF strains used.

Our study revealed that the survival rate of *S. frugiperda* was significantly lower when they fed on leaves inoculated with fungal endophytes, compared to those fed on untreated leaves. Distinctive symptoms were observed in the dead larvae, characterized by their shrunken and rigid mummy-like appearance. The larvae’s bodies were covered with fungal mycelia and changed color to either white or green, depending on the fungal species that infect and demise them. Larvae that consumed leaves contaminated with *B. bassiana* and *M. anisopliae* resulted in cadavers exhibiting white and green colors, respectively.

Additionally, all life table parameters, including survival rate, life expectancy, reproductive values, fecundity, and maternity rate were adversely affected by the application of fungal endophytes compared with the control. Previous studies have also reported the negative impact of endophytic EPF on the life history parameters of insect pests [56,57,58]. Mortality rates of insect pests using EPF depend on various factors such as the larval developmental stage [32], fungal strain [59], and inoculation method [51]. For example, Ramirez-Rodriguez et al. [60] reported that *B. bassiana* isolates from soil caused 98.3% mortality of 3rd instar larvae of *S. frugiperda*, whereas the same strain isolated from endophytically colonized maize plants caused 75% mortality.

Our study demonstrates that the endophytic colonization of plants with EPF can have a negative impact on the population of *S. frugiperda*. Our results showed that larvae and pupae had a prolonged developmental period, and both fecundity and survival rates were reduced. Previous studies have also reported the effectiveness of various EPFs, such as *B. bassiana* and *M. anisopliae*, in suppressing insect pests [14,52,61,62,63,64].

## 5. Conclusions

The results of our study indicated that endophytic fungi, when applied to maize plants, had a negative impact on the population of *S. frugiperda*. We observed a reduction in key life history parameters such as developmental period, reproduction potential, and survival rate of the pest. These findings suggest that both EPFs have potential as endophytes in integrated pest management (IPM) strategies to protect maize plants against this destructive pest. It is worth noting, however, that our study was conducted under controlled conditions and further research is needed to confirm the EPFs’ effectiveness in field conditions.

## Figures and Tables

**Figure 1 microorganisms-11-01067-f001:**
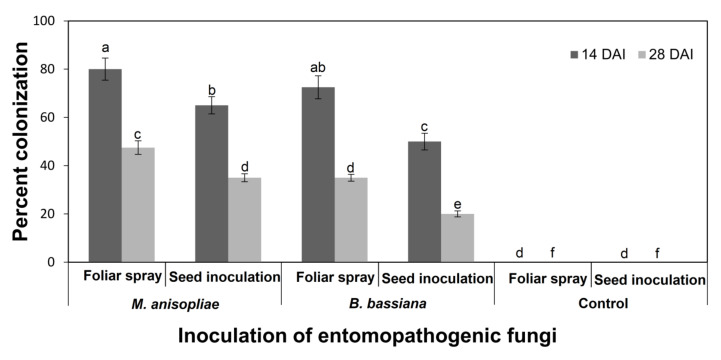
Percent colonization (means ± SE) of entomopathogenic fungi on maize with two inoculation methods at 14 and 28 days after inoculation (DAI) (LSD test after three-way ANOVA). Different letters above bars indicate significantly different means.

**Figure 2 microorganisms-11-01067-f002:**
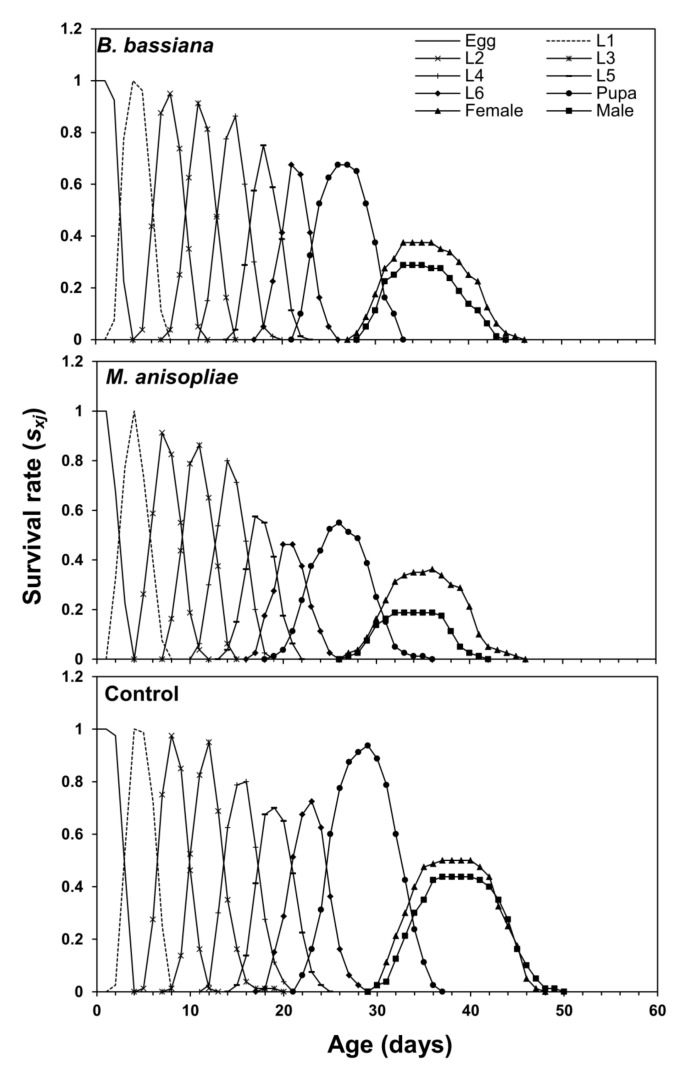
Age-stage-specific survival rate (*s_xj_*) of *Spodoptera frugiperda* fed on endophytic colonized and non-colonized plants.

**Figure 3 microorganisms-11-01067-f003:**
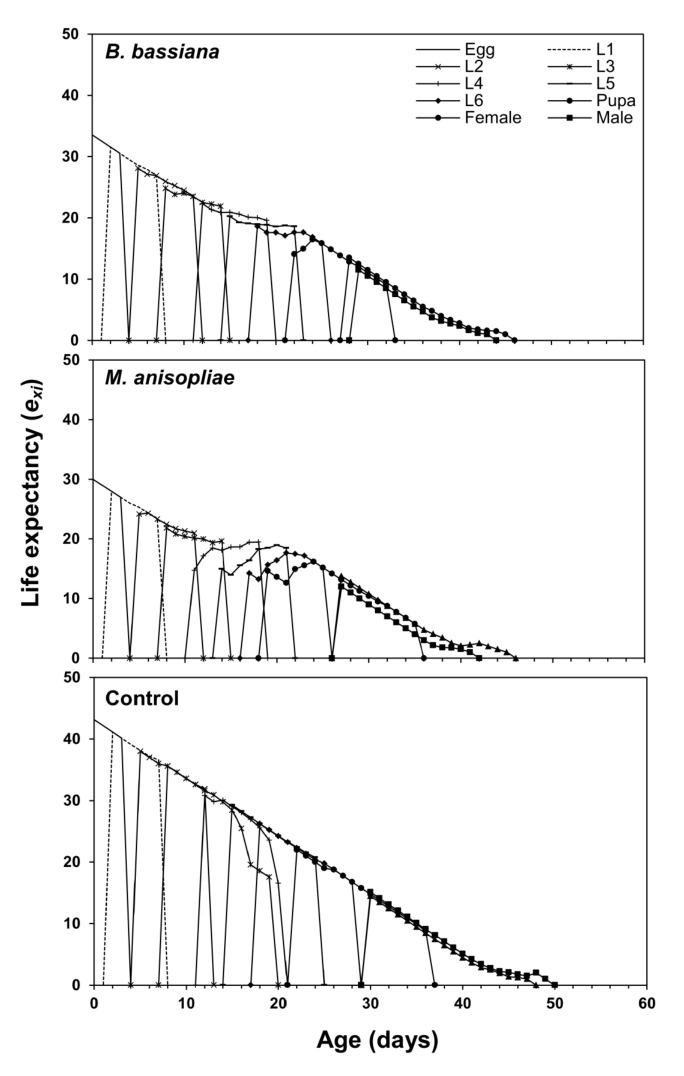
Age-stage-specific life expectancy (*e_xj_*) of *Spodoptera frugiperda* fed on endophytic colonized and non-colonized plants.

**Figure 4 microorganisms-11-01067-f004:**
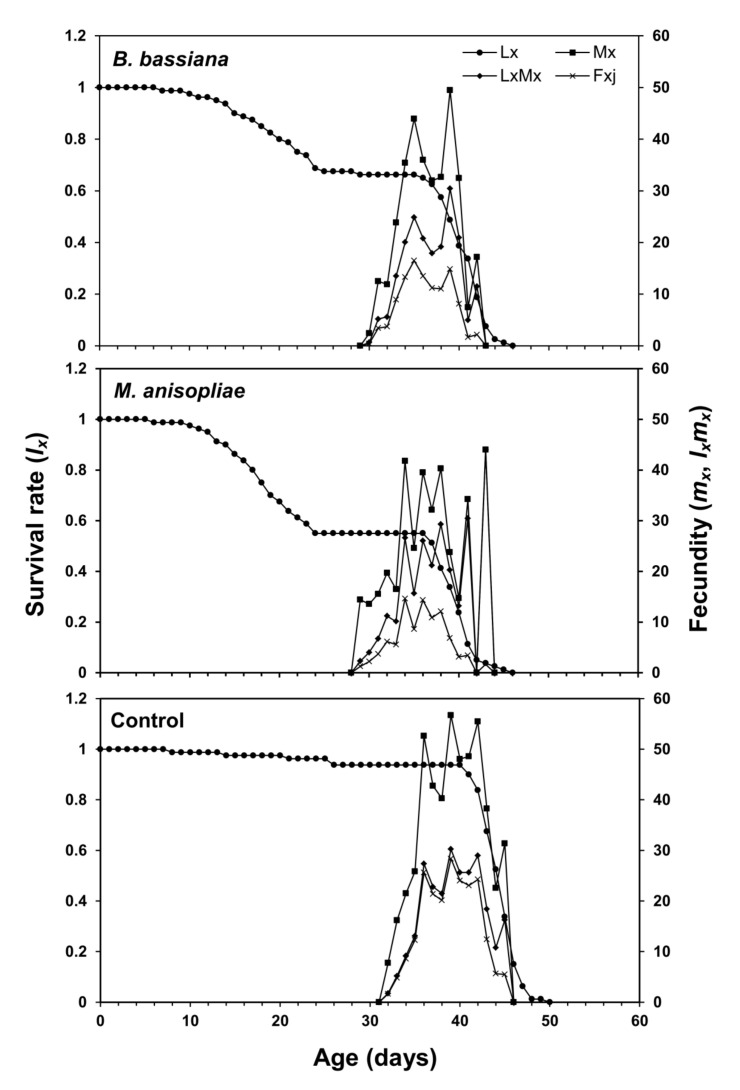
Age-stage-specific survival rate (*l_x_*), age-stage specific fecundity (*f_xj_*), age-specific fecundity (*m_x_*) and age-specific maternity (*l_x_m_x_*) of *Spodoptera frugiperda* fed on endophytic colonized and non-colonized plants.

**Figure 5 microorganisms-11-01067-f005:**
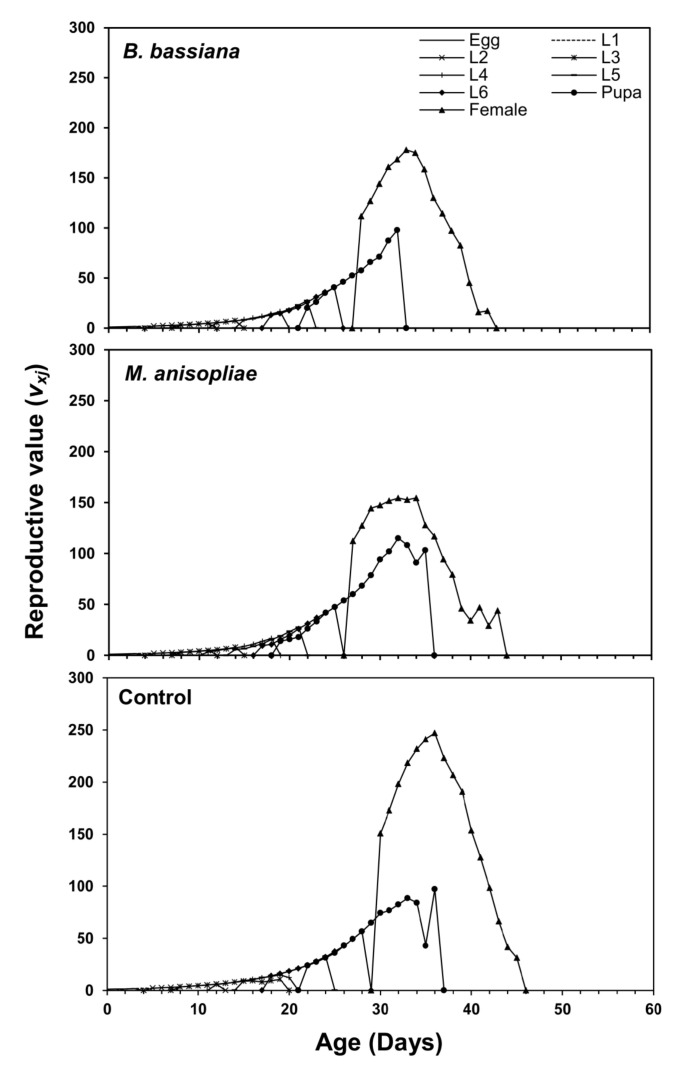
Age-stage-specific reproductive value (*v_xj_*) of *Spodoptera frugiperda* fed on endophytic colonized and non-colonized plants.

**Table 1 microorganisms-11-01067-t001:** Development period (average no. of days) of *Spodoptera frugiperda* fed on maize plants colonized with *Beauveria bassiana* and *Metarhizium anisopliae* in comparison with untreated plants.

Life Stages	*n*	*B. bassiana*	*n*	*M. anisopliae*	*n*	Control
Egg incubation	80	3.15 ± 0.059 b	80	2.90 ± 0.083 c	80	3.44 ± 0.061 a
L1	80	3.49 ± 0.056 a	80	3.31 ± 0.055 b	80	3.52 ± 0.056 a
L2	79	3.47 ± 0.065 a	79	3.39 ± 0.058 a	79	3.53± 0.074 a
L3	76	3.41 ± 0.065 b	77	3.77 ± 0.086 a	79	3.44 ± 0.057 b
L4	71	3.56 ± 0.069 a	69	3.49 ± 0.064 a	77	3.62 ± 0.064 a
L5	65	3.29 ± 0.065 b	59	3.51 ± 0.086 a	77	2.98 ± 0.062 c
L6	60	3.42 ± 0.083 b	51	3.74 ± 0.068 a	77	3.18 ± 0.118 b
Pupa	54	8.20 ± 0.081 a	44	7.43 ± 0.110 b	75	6.91± 0.109 c
Adult Longevity	53	10.3 ± 0.237 b	44	9.73 ± 0.135 c	75	11.4 ± 0.086 a
Male adult Longevity	23	9.74 ± 0.310 b	15	9.20 ± 0.170 b	35	11.4 ± 0.130 a
Female adult Longevity	30	10.8 ± 0.320 b	29	10.0 ± 0.160 b	40	11.4 ± 0.110 a

SE was estimated by Bootstrapping (100,000 replications), and L1–L6 indicates the larval instar. *n* = shows the number of individuals; means sharing similar letters are not significantly different determined using the paired bootstrap test (*p* < 0.05); L1–L6 shows the larval instars.

**Table 2 microorganisms-11-01067-t002:** Comparison of reproductive and life table parameters (mean ± SE) of *Spodoptera frugiperda* fed on maize plants colonized with *Beauveria bassiana* and *Metarhizium anisopliae* in comparison with untreated plants.

Parameters	*B. bassiana*	*M. anisopliae*	Control
APOP	2.67 ± 0.110 a	2.55 ± 0.090 ab	2.35 ± 0.080 b
TPOP	33.3 ± 0.270 b	33.3 ± 0.400 b	35.3 ± 0.280 a
Oviposition days	3.80 ± 0.120 b	3.62 ± 0.090 b	4.58 ± 0.090 a
Fecundity	290.1 ± 9.870 b	260.0 ± 8.030 c	435.6 ± 10.930 a
*Ro* (offspring individual^−1^)	108.7 ± 16.11 b	94.30 ± 14.31 c	217.8 ± 24.99 a
*T* (d)	36.8 ± 0.300 b	36.3 ± 0.430 b	39.3 ± 0.280 a
*r* (d^−1^)	0.127 ± 0.004 b	0.125 ± 0.004 b	0.133 ± 0.003 a
*λ* (d^−1^)	1.135 ± 0.004 b	1.133 ± 0.005 b	1.146 ± 0.003 a

SE was estimated by bootstrapping (100,000). Whereas *R*_0_ = Net reproductive rate, *r* = Intrinsic rate of increase, *λ* = Finite rate of increase, *T* = Mean generation time; means sharing similar letters are not significantly different as determined using the paired bootstrap test (*p* < 0.05).

## Data Availability

All data analyzed in this study are included in this article.

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
