# Peer review of "Endophytic Colonization by Beauveria bassiana and Metarhizium anisopliae in Maize Plants Affects the Fitness of Spodoptera frugiperda (Lepidoptera: Noctuidae)"

_microorganisms, 2023, doi:10.3390/microorganisms11041067_

Round 1

Reviewer 1 Report

Line110:  When the maize seedlings were three weeks old, they were sprayed using a hand sprayer with an average of 3 mL of spore suspensions ……., but how many leave of maize at this stage of maize.

Line126: ten plants were selected, and one leaf was taken from each plant for each treatment. How about the leaf position, and all the sampled leaf should be the same position in each maize plant.

2. Materials and Method : B. bassiana and M. anisopliae afftected the life table parameters of Spodoptera frugiperda by the Endophytic colonization B. bassiana and M. anisopliae in maize plants, authors evaluated the Endophytic effects of entomopathogenic fungi on life table parameters of S. frugiperda, but how about the B. bassiana and M. anisopliae in those larve of S. frugiperda after feeding the Endophytic colonized maize plants. If not do so, it may be discussed in the discussion.

Author Response

Reviewer 1

Thanks for your valuable comments. All changes were done with red color

Line110:  When the maize seedlings were three weeks old, they were sprayed using a hand sprayer with an average of 3 mL of spore suspensions ……., but how many leave of maize at this stage of maize.

Ans: When the maize seedlings were three weeks old, the plants were at the growth stage BBCH 15 (5 leaves unfolded) (Meier 2001). The information has been added in the M&Ms section.

Line126: ten plants were selected, and one leaf was taken from each plant for each treatment. How about the leaf position, and all the sampled leaf should be the same position in each maize plant.

Ans: The fourth true leaf was taken from each plant. The information has been added in the M&Ms section.

  1. Materials and Method : B. bassianaand M. anisopliae afftected the life table parameters of Spodoptera frugiperdaby the Endophytic colonization B. bassiana and M. anisopliae in maize plants, authors evaluated the Endophytic effects of entomopathogenic fungi on life table parameters of S. frugiperda, but how about the B. bassiana and M. anisopliae in those larve of S. frugiperda after feeding the Endophytic colonized maize plants. If not do so, it may be discussed in the discussion.

Ans: The authors are thankful to reviewer for this useful comment. After exposure to larvae with treated leaves, distinctive symptoms were observed in the dead larvae, characterized by their shrunken and rigid mummy-like appearance. The larvae's bodies were covered with fungal mycelia and changed color to either white or green, depending on the fungal species that infect and demise them. Larvae that consumed leaves contaminated with B. bassiana and M. anisopliae resulted in cadavers exhibiting white and green colors, respectively. The information has been added in the Discussion section.

Reviewer 2 Report

The paper appears well planned, relatively well written, and represents a good contribution to science literature. This information warrants publication; however, this paper requires moderate revision.

The description of the experiments is not clear. I invite the authors to carefully revise the material and methods section paying attention to mention how many times the experiments were repeated and the experimental design. At the moment, it is not possible to understand if the results reported in the tables derive from a single experiment or they are a pool of two or more experiments. It is highly important to report this information in the material and methods section and in the table legends. As well as the analysis of variance of the experimental design.

Author Response

Reviewer 2

Thanks for your valuable comments. All changes were done with red color

The paper appears well planned, relatively well written, and represents a good contribution to science literature. This information warrants publication; however, this paper requires moderate revision.

Ans; The authors are thankful to reviewer for such a positive feedback.

The description of the experiments is not clear. I invite the authors to carefully revise the material and methods section paying attention to mention how many times the experiments were repeated and the experimental design. At the moment, it is not possible to understand if the results reported in the tables derive from a single experiment or they are a pool of two or more experiments. It is highly important to report this information in the material and methods section and in the table legends. As well as the analysis of variance of the experimental design.

Actually it was the life table study and eighty eggs were selected for each treatment to study the life history and each egg was considered as one replication. We followed the methodology from previous published articles related to life table theory and some references are given below. In complete life-table experiments, a group of individuals of the same age observe from the beginning of their egg stage until the death of all adults. During insect development, all phenological events (i.e., development time of immature life stages, mortality, longevity, and fecundity) record at constant intervals of generally 1 day. The phenological events monitor until the development to the next stage or death of all individuals in the cohort. The life table raw data of S. frugiperda were analyzed according to the age-stage, two-sex life table with the computer program TWOSEX-MSChart that is specific to life table studies. Further we have improved the information about the analysis of variance and the experimental design in M&Ms.  

  1. Ali, et al (2020). Insects, 11(5), 275. https://doi.org/10.3390/insects11050275
  2. Sun et al., (2020). Insects 2020, 11(3), 181; https://doi.org/10.3390/insects11030181
  3. Chen et al., (2023). Insects 2023, 14(4), 329; https://doi.org/10.3390/insects14040329

Reviewer 3 Report

My first and, in many ways, greatest difficultly with this manuscript is its potential inclusion in a special issue on advances in research on ancient terrestrial fungi. In a phylogenetic sense it is virtually impossible for me to accept Beauveria, Metarhizium, or any other cordycipitoid entomopathogens as 'ancient' terrestrial fungi. If the guest editors and journal were to change the thrust of this special issue to entomopathogenic fungi in general, then I would have no concerns about its appropriateness for such a special issue.

I have provided a marked copy of this manuscript on which my added comments discuss a series of scientific and editorial concerns about this manuscript.

On the scientific side, here is a summary of those comments:
• The genus Isaria mentioned several times, not exclusively in connection
   with cited references) is no longer accepted as a valid genus under
   the current phylogenetically based rules of fungal nomenclature.
• It is necessary to state the strength of the sodium hypochlorite bleach
   solutions noted several times in the Materials and Methods.
• I am not convinced that the experiments were satisfactorily replicated.
   To use, for example, 20 plants for the foliar applications inoculated at the
   same time with the same fungal preparations appear to this reviewer to be
   pseudoreplications rather than true replications (which would require 
   performing the experiments multiple times using new materials and on
   different dates).
• There is no apparent mention or discussion about the data in Table 1
   showing that the incubation times for eggs exposed to either fungus were
   shorter than for the controls. Nothing was mentioned about possible fungal
   infections causing mortality of the treated eggs.

It seems problematic that the authors included no comparative data of how infective the initial conidial powders of the fungal isolates used here were against the various stages of Spodoptera frugiperda. The lack of such data then mean there is no good context for comparing the effectiveness of potential seed or foliar applications for pest control as corn endophytes may be versus more standard foliar or soil applications of these fungi against this target pest.

The other comments that were added to the manuscript are of a more editorial nature and should be easy for the authors to address.

Author Response

Reviewer 3

Thanks for your valuable comments. All changes were done with red color

The genus Isaria mentioned several times, not exclusively in connection with cited references) is no longer accepted as a valid genus under the current phylogenetically based rules of fungal nomenclature.

Ans: The information related to genus Isaria has been deleted from the manuscript.

It is necessary to state the strength of the sodium hypochlorite bleach solutions noted several times in the Materials and Methods.

Ans: The strength of sodium hypochlorite was 1.0%. The information has been added.

I am not convinced that the experiments were satisfactorily replicated. To use, for example, 20 plants for the foliar applications inoculated at the same time with the same fungal preparations appear to this reviewer to be pseudoreplications rather than true replications (which would require performing the experiments multiple times using new materials and on different dates).

Ans: Thank you for bringing this to our attention. We apologize for the mistake regarding the replication in our article. Actually, the experiment was repeated four times in case of both inoculation methods. For each treatment we selected 20 plants randomly (5 from each replication). Independent batches of plants and EPF were used in each treatment. The methodology was followed by Zhang et al., (2021). https://doi.org/10.21203/rs.3.rs-726671/v1 The information has been updated in the M&Ms section.

There is no apparent mention or discussion about the data in Table 1 showing that the incubation times for eggs exposed to either fungus were shorter than for the controls. Nothing was mentioned about possible fungal infections causing mortality of the treated eggs.

Ans: The eggs were not exposed to EPFs. The inoculated leaves were fed to 2-days old larvae and then the effect was determined throughout the life table. That’s why it is not discussed in the article about the egg incubation period.

It seems problematic that the authors included no comparative data of how infective the initial conidial powders of the fungal isolates used here were against the various stages of Spodoptera frugiperda. The lack of such data then mean there is no good context for comparing the effectiveness of potential seed or foliar applications for pest control as corn endophytes may be versus more standard foliar or soil applications of these fungi against this target pest.

Ans: Thank you for your comment. We understand the importance of comparative data regarding the initial infectivity of the fungal isolates against Spodoptera frugiperda. A germination test was performed for both fungi to evaluate the viability of conidial spores. The conidial suspensions with ≥ 90% germination were used for plant inoculation. In this study, our main focus was to determine the endophyte effects of these EPF on life table parameters of Spodoptera frugiperda. And the comparative effect of two endophytic EPF is give in the article in terms of developmental period of each stage, survival and fecundity rate and age-stage specific parameters as well. Further the conidial powder of both fungi has been determined earlier against Spodoptera litura and frugiperda. Our published articles are here;

  1. Mubeen, et al., (2022). Effect of Metarhizium anisopliae on the nutritional physiology of the fall armyworm, Spodoptera frugiperda (JE Smith)(Lepidoptera: Noctuidae). Egyptian Journal of Biological Pest Control, 32(1), 1-5.
  2. Ullah et al., (2019). Effects of Entomopathogenic Fungi on the Biology of Spodoptera litura (Lepidoptera: Noctuidae) and its Reduviid Predator, Rhynocoris marginatus (Heteroptera: Reduviidae). Int. J. Insect Sci. 11: 1-7.

The other comments that were added to the manuscript are of a more editorial nature and should be easy for the authors to address.

Incomplete address without adding the country

Ans: The changes have been made as suggested

EPF is here defined as a plural term, and it is so defined in the body of the manuscript. It is extraneous, then, here in the abstract or elsewhere in the paper to refer to 'EPFs' as a plural. Please do not do so!

Ans: The changes have been made as suggested

BETTER: 14 days after inoculation.

Ans: The changes have been made as suggested

Slower development to what? The times given are to the emergence from the pupae of the adults? This needs to be specified.

Ans: Here we mentioned the results for developmental period of larvae in which slower development was reported as compared to control groups. It is mentioned in the statement that its about larvae.

strong flying [no hyphen]

Ans: The changes have been made as suggested

Isaria is no longer accepted as a valid genus under the new rules of fungal nomenclature adopted in 2011. Almost all species formerly placed in isaria are now classified as species of Cordyceps.

Ans: The information related this Isaria genus has been deleted throughout the article

What was the strength of the sodium hypochlorite bleach solution? This needs to be specified in all mentions of its use here in the Materials and Methods.

Ans; The information has been updated

medium [media is the plural form of this noun but you use only a single medium; this is too common an error that really must be avoided]

Ans; The changes have been made as suggested.

showing fungal growth. In the denominator, add the space in 'no. of plated'.

Ans; The changes have been made as suggested.

There does not seem to be any discussion of this peculiar finding that the incubation times for eggs exposed to both fungi were shorter than for the controls. Neither does there appear to be any discussion of whether any egg mortality was caused by fungal infections.

Ans; Already answered above

Musa needs to be italicized

Ans; The changes have been made as suggested.

Round 2

Reviewer 3 Report

I appreciate the authors' rapid responses to the concerns I raised about the previous version. I believe that very nearly all of those responses have been appropriate and meet the immediate needs for this manuscript's improvement. The only concern I felt was not addressed appropriately was the one asking about the comparative efficacy of the fungal strains used in direct applications to S. frugiperda (for which the authors provided two references to their own work on that exact subject) versus the administration of the fungi as corn endophytes. I honestly believe that the paper would be strengthened by mentioning these uncited references (and any other relevant citations by other authors using Beauveria or Metarhizium against S. frugiperda) in the Introduction and/or Discussion.

Author Response

Response to Reviewer-3-R2

Comment:

I appreciate the authors' rapid responses to the concerns I raised about the previous version. I believe that very nearly all of those responses have been appropriate and meet the immediate needs for this manuscript's improvement. The only concern I felt was not addressed appropriately was the one asking about the comparative efficacy of the fungal strains used in direct applications to S. frugiperda (for which the authors provided two references to their own work on that exact subject) versus the administration of the fungi as corn endophytes. I honestly believe that the paper would be strengthened by mentioning these uncited references (and any other relevant citations by other authors using Beauveria or Metarhizium against S. frugiperda) in the Introduction and/or Discussion.

Response:

Thanks for your comment. We agree with this comment and we have added in lines [284-287] the following (in red color) with five new references, then, we adjusted the references order.

In general, many of recent investigations stated that numerous B. bassiana and M. anisopliae isolates have shown the high efficiency of these fungi in infection and control of S. frugiperda larvae [34-38].

  1. Mwamburi, L.A. Endophytic fungi, Beauveria bassianaand Metarhizium anisopliae, confer control of the fall armyworm, Spodoptera frugiperda (J. E. Smith) (Lepidoptera: Noctuidae), in two tomato varieties.  J. Biol. Pest Control. 2021, 31, 7.
  2. Montecalvo, M.P.; Navasero, M.M. Comparative virulence of Beauveria bassiana (Bals.) Vuill. AND Metarhizium anisopliae (Metchnikoff) Sorokin TO Spodoptera frugiperda (J.E. Smith) (Lepidoptera: Noctuidae). ISSAAS. 2021, 27(1), 15-26.
  3. Idrees, A.; Afzal, A.; Qadir, Z.A.; Li, J. Bioassays of Beauveria bassianaIsolates against the Fall Armyworm, Spodoptera frugiperda. J. Fungi. 2022, 8(7), 717.
  4. Idrees, A.; Afzal, A.; Qadir, Z.A.; Li, J. Virulence of entomopathogenic fungi against fall armyworm, Spodoptera frugiperda (Lepidoptera: Noctuidae) under laboratory conditions. Physiol. 2023, 14, 1107434.
  5. Fazlullah; Shahid, H.; Muzammil, F.; Aslam, M.N.; Zada, N. Insecticidal potential of eco-friendly mycoinsecticides for the management of fall armyworm (Spodoptera frugiperda) under in vitro conditions. J. Agric. Sci. 2023, 29(1), 124-130.

We hope that will be in agreement with your comment.
